# Generating brain-wide connectome using synthetic axonal morphologies

Remy Petkantchin [1] ✉, Adrien Berchet [1], Hanchuan Peng [2,3], Henry Markram[1] & Lida Kanari[1]

Recent experimental advancements, including electron microscopy reconstructions, have produced detailed connectivity data for local brain regions. On the other hand, for inter-regional connectivity, large-scale imaging techniques such as MRI are best suited to provide insights. However, understanding the relationship between local and long-range connectivity is essential for studying both healthy and pathological conditions of the brain. Leveraging a dataset of whole-brain axonal reconstructions, we present a technique to predict whole-brain connectivity at single cell level for pyramidal cells in the cortex by generating detailed whole-brain axonal morphologies from sparse experimental data. The computationally generated axons accurately reproduce the local and global morphological properties of experimental reconstructions. Furthermore, the computationally synthesized axons generate large-scale inter-regional connectivity, defining the projectome and the connectome of the brain, thereby enabling the in silico experimentation of large brain regions.

Computational simulation and synthesis of the brain are essential for studying the structural and functional complexity of the brain[1,2]. While significant progress has been made in synthesizing dendritic morphologies and neuronal networks, the synthesis of axonal trees remains a critical challenge. Biophysically accurate models simulate neural growth by incorporating molecular mechanisms of neuronal development[3], although their computational intensity limits scalability. Phenomenological models, based on mathematical principles[4,5] or statistical sampling from morphological datasets[6,7], provide computationally efficient alternatives but often require manual tuning and lack generalizability. Recently, we introduced a computationally efficient method for synthesizing biologically realistic dendrites[8] based on the topological morphology descriptor[9,10], and a similar approach has been applied to astrocytes[11]. However, generative models predominantly focus on dendritic synthesis and do not account for brain geometry and preferential axon targeting, therefore disregarding their impact to large-scale connectivity.

Unlike dendrites, which are generally restricted to a single brain region despite their environmental sensitivity[12–14], axons extend across brain regions, spanning much larger scales[15]. This highlights the necessity of generating axons that are spatially embedded within the brain[16]. Although previous modeling efforts have explored aspects such as axon guidance[17], trade-offs between material and conduction delay[18], and fiber arrangement[19], only the ROOTS model[20] has addressed the challenge of synthesizing detailed axonal morphologies. However, the ROOTS model is limited to a single brain region, the rat dentate gyrus, and does not extend across multiple brain regions, limiting its broader applicability to long-range axons (LRAs).

Brain-wide LRAs are the primary drivers of inter-regional connectivity[21] and plasticity[22]. Synthesizing LRAs presents a significant challenge as it requires replicating their projection patterns and morphometric features while also considering brain topology. In ref. 23, we introduced an algorithm for synthesizing realistic axonal morphologies, addressing a key limitation of previous models. This algorithm generates LRAs based on the terminal points of axonal reconstructions, accurately reproducing the morphological properties of reconstructed LRAs. It solves the challenge of efficiently connecting neuronal somata to multiple targets while preserving the statistical

[1]Blue Brain Project, EPFL, Geneva, Switzerland. [2]New Cornerstone Science Laboratory, Institute for Brain and Intelligence, Fudan University, Shanghai, China. [3]Shanghai Academy of Natural Sciences (SANS), Fudan University, Shanghai, China. ✉e-mail: remy.pet@gmail.com

properties of axonal shapes. However, it does not explicitly address the problem of axon targeting as a generic biological property of LRAs, which is essential for integrating synthetic axons into large-scale connectivity models. Without an accurate definition of brain-region targeting, this process cannot generalize across multiple regions, preventing the generation of meaningful connectivity maps.

In this work, we present a complete pipeline for synthesizing brain-wide LRAs and establishing single-cell resolution brain-wide connectivity. To achieve this, we address the issue of input variability by generating a dataset of 800 new LRAs using techniques developed in ref. 24, focusing on cortical axonal targeting. Combined with publicly available LRA datasets[24,25], this targeted dataset allows us to identify the targeting behavior of the subregions of cortical LRA populations. To generalize axon targeting from available biological inputs, we employ an unsupervised Gaussian Mixture Model (GMM) clustering approach to group input morphologies based on their projection patterns. This approach ensures that biological projection patterns and proportions are accurately reproduced, reflecting the targeting behavior of axons during development. The resulting clusters serve as inputs for our long-range axon synthesis algorithm[23], extending single-cell LRA synthesis to a population-wide model that preserves target-source relationships across brain regions. The synthesized axons traverse the brain, reaching their designated target regions according to their cluster assignments, with a preference for following fiber tracts to enhance biological realism. This data-driven approach enables the generation of accurate connectivity maps, providing a crucial step toward full-brain simulations.

Successfully synthesizing LRAs while capturing their targeting properties allows us to construct synthetic connections between neurons across the brain, bridging the gap between the connectome, which encodes individual neuronal connections, and the projectome[26], which encodes region-to-region connectivity. Once this connectivity is established, it becomes possible to extend local simulations[2] to circuits spanning multiple brain regions, advancing toward biologically realistic whole-brain simulations.

This species-agnostic workflow is applied to a dataset of brain-wide axons from the mouse brain containing previously published[24,25] and new data (with same methods as ref. 24), producing the a simulation-ready synthetic mouse brain with LRAs. In this work, we demonstrate our algorithms on excitatory cells of the isocortex, specifically emphasizing layer five of the primary motor area due to the larger data availability in this region. LRAs of pyramidal cells are computationally synthesized, while local axons are grafted onto interneurons, following the approach used in our previous work[2,8]. We validate the synthesized axons by comparing their morphological, topological, and targeting properties, demonstrating statistical similarity to experimental reconstructions. We furthermore validate our results by comparing the projection ratio to targeted regions against experimental data[27] that were not used as input to test the emergent properties of our algorithm. Finally, connectivity, computed with a method based on spatial proximity (see "Methods" Creating connections), emerges statistically similar to the input biological axons, which concludes the validation.

## Results

### Clustering

We show in Fig. 1 an example of clustering of 59 axonal morphologies from the MouseLight dataset[25] that originate from the presubiculum region of the mouse brain. In Fig. 1A, B, C, each column represents an axon, and each row is a region where it terminates. Axons are grouped into the GMM clusters (top row) and based on their parent brain area (left column). We imposed a number of five clusters for these axons to echo Wheeler et al.[28], who found five clusters using unsupervised hierarchical clustering of axonal projections originating in the presubiculum. In Fig. 1A, we clustered the axons based on the axonal path

lengths in the regions and sampled the lengths of the same number of axons from these clusters in Fig. 1B. These sampled axons are *virtual* in the sense that we did not synthesize them; we only sampled their lengths to verify the clustering. Some noise can be observed because a scalar variance was selected for each cluster by the Bayesian Information Criterion (BIC) optimization. Nevertheless, the main projections could be reproduced for each cluster. In Fig. 1C, we used the number of terminals in regions as clustering feature. The clusters are defined slightly differently due to the unsupervised nature of the clustering. However, we show in Fig. 1D that both features can be used equivalently for clustering. In the latter figure, each point corresponds to a target region, for which we computed the sum of axonal path length versus the number of terminals, for all axons of ref. 25. A linear relationship was found, showing that one feature is proportional to the other.

### Synthesis in the mouse brain

In this section, we showcase two cases of LRA synthesis: the first originating from the primary motor area layer 5 (MOp5) and the second from multiple regions in the isocortex. We study the former in detail and use the latter as a window into the possibilities of the present methodology. In both cases, we synthesized 54401 cells from the isocortex (~1% of the actual density). Cells for which LRAs were not synthesized have grafted local axons, as done in ref. 8. Local axons typically have a total path length of about 20 mm, whereas LRAs can go above 50 mm and up to 400 mm (distributions of lengths in Supplementary Fig. S4B).

**MOp5 pyramidal cells.** From the 54401 synthesized cells of the isocortex, 1695 were pyramidal cells from the MOp5, for which we synthesized LRAs in this section.

In Fig. 2, we offer a visualization of the 65 biological morphologies from the dataset that originated in the MOp5 (in blue) and 65 randomly picked synthesized morphologies with LRAs (in red). In Fig. 2A, we projected the reconstructions and the synthesized morphologies on a flat map of the isocortex, as described in ref. 29. In Fig. 2B, C, we used the `Brayns` tool[30] to visualize the mouse brain in three dimensions. We qualitatively see that synthesized axons follow the same paths and terminate in the same regions as the biological reconstructions. We proceed with a quantitative comparison of the two populations.

**Morphometrics**: In Fig. 3A, B, we show a set of morphometrical features for the tufts (Fig. 3A) and trunks (Fig. 3B). Since there were more biological axons in the left (46) versus right (19) hemisphere, for the subsequent comparisons, we considered only a subset (362) of synthesized axons from the right hemisphere, to keep the same proportions. The distributions were normalized as in ref. 8, where the value for an axon $a$ was centered to the median of the reconstructed population $\tilde{P}_r$, and divided by the standard deviation of the reconstructed population $\sigma(P_r)$:

$$V_{\text{norm}}(a) = \frac{\tilde{P}_r - V(a)}{\sigma(P_r)}. \tag{1}$$

The synthesized and clustered tufts morphometrical features are in good agreement. The tufts were synthesized with the dendrites synthesis algorithm, which has already proved to reproduce well morphometrics of input dendritic trees[8]. This shows that dendrites and tufts can be mathematically described with the same topological method. Morphometrics of the axon trunks were found to match reasonably well with biological inputs when they were mimicked, as shown in ref. 23. However, with the new selection of target points presented in this work, many morphometrics, such as the number of leaves, angles, and those related to distances and lengths, are subject to change. This resulted in overall longer section lengths and path distances, slightly increased remote bifurcation angles, and slightly

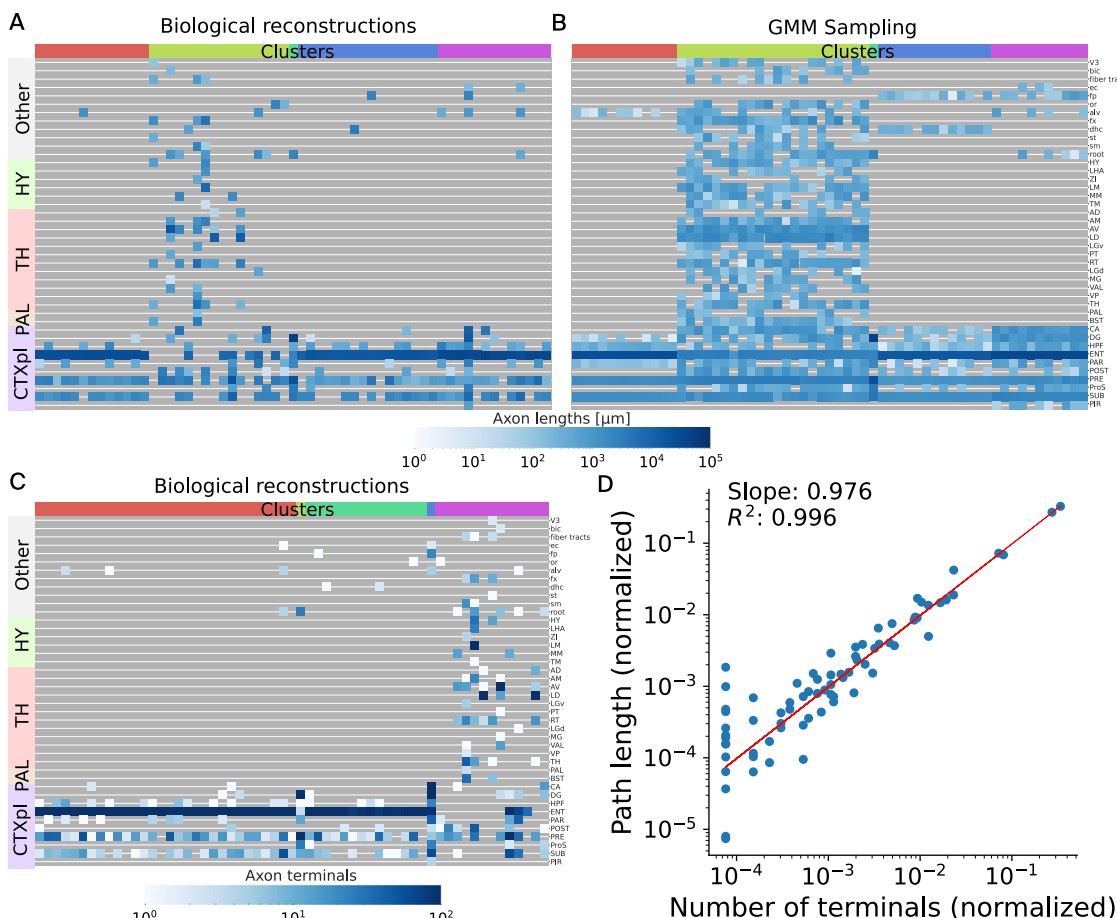

**Fig. 1 | Clustering of 59 axonal morphologies from ref. 25 with somata in the presubiculum.** A total number of five clusters is imposed here because[28] reported five clusters in their clustering. **A** Clustering based on axon lengths in target regions where the axons terminate. **B** Projections of 59 virtual axons sampled with the GMMs defined by axon lengths (**A**). **C** Clustering based on the number of terminals in the regions where the axons terminate. **D** Using the number of terminals or axon lengths in target regions as feature vector for clustering was found to be equivalent. A linear relationship was found between number of terminals and path lengths of axons in the regions. GMM Gaussian mixture model, CTXpl Cortical plate, PAL Pallidum, HY Hypothalamus, TH Thalamus.

fewer leaves and branches. We finally compared the path lengths of the axons in targeted regions in Fig. 3C, D. Fig. 3C focuses on the layers of the MOp (meso-scale), and Fig. 3D on the set of regions highlighted in Fig. S3 (macro-scale): the caudoputamen (CP), the primary and secondary motor areas (MOp, MOs), the pontine gray region (PG), the spinal nucleus of the trigeminal interpolar part (SPVI), and the primary and secondary somatosensory areas (SSp, SSs). The top subplot of Fig. 3C, D shows the proportions of reconstructed (blue) and synthesized (red) axons terminating in the regions; in the middle subplot, the distribution of axon lengths, and finally, in the bottom subplot, the total lengths of all axons, normalized by the number of axons in the populations. The stars in the middle subplots show strong significance (**) and non-randomness (*) when the Maximum Visible Spread (MVS) score between the populations is <0.1, respectively <0.5 (see section "Long-range axon synthesis" and ref. 8 for a definition of the MVS). Overall, we can see that the terminating number of axons and their path lengths in the targets were accurately replicated.

**Targeting**: We now compare the location of clustered and synthesized tufts in Fig. 3E, by looking at the location of the common ancestors of the tufts computed in the reconstructed axons (top row) and placed in the synthesized axons (bottom row). We only show a set of brain regions, which were reported to be targeted by 42 MOp5 axons in the dataset analyzed in ref. 27: the cortical subplate (CTXsp), isocortex, medulla (MY), PG, thalamus (TH), hypothalamus (HY), midbrain (MB), olfactory areas (OLF), pallidum (PAL), and striatum (STR). The exploded slices of the pie charts show targets in the

contralateral hemisphere, whereas the normal slices are in the ipsilateral hemisphere. We see that the proportions of targets clustered and synthesized and the number of axons starting from each hemisphere were statistically well reproduced. In Fig. 3F, we compared the projection ratio from a different set of biological reconstructions from ref. 27 and our synthesized axons. The projections ratio is defined as the proportion of axons targeting each region. Even though we used a different source for biological inputs in this work, the synthesized axons primarily targeted the same areas, and their projections ratio was not far from ref. 27, with the highest difference of 10.4% in PG, and around 5 to 7% in CTXsp, HY, STR, and MB.

**Connectivity**: Finally, we computed the connectivity of the MOp5 pyramid cells and show the results in Fig. 4. We chose to display here the connections in all the subregions of the isocortex: the anterior cingulate area (ACA), frontal pole of the cerebral cortex (FRP), gustatory areas (GU), infralimbic area (ILA), MOp, MOs, orbital area (ORB), prelimbic area (PL), retrosplenial area (RSP), SSp, SSs, auditory areas (AUD), ectorhinal area (ECT), perirhinal area (PERI), temporal association areas (TEa), visceral area (VISC), visual areas (VIS), agranular insular area (AI) and posterior parietal association areas (PTLp).

First, we compared in Fig. 4A, B the proportion of outgoing connections of the 65 biological reconstructions from MOp5 (A) and the synthesized axons of the MOp5 pyramidal cells, with the same proportions in both hemispheres (B). The arrows represent the direction of efferent axo-dendritic synaptic connections, and their size is proportional to the number of connections. We did not show in this figure

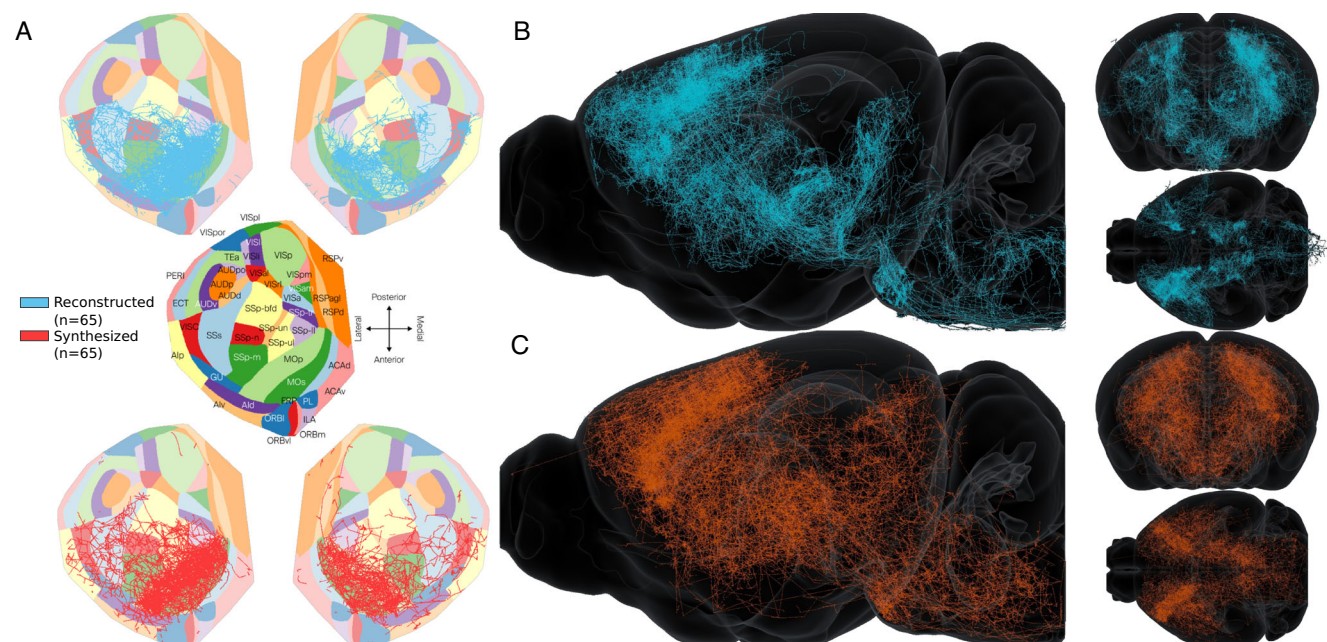

**Fig. 2 | Comparison of 65 reconstructed (blue) and 65 of the 1695 synthesized axons (red) of MOp5 pyramidal cells. A** Flat map projection in the isocortex of the reconstructed and 65 randomly selected synthesized axons. **B** The 65 reconstructed axons in the mouse brain atlas, in lateral (left), frontal and top views (right). Some axons go out of the boundaries because reconstruction techniques or brain annotation space might vary between morphologies. **C** 65 of the synthesized axons in the mouse brain atlas, lateral (left), frontal and top views (right).

the connections within MOp because all the synthesized cells in the MOp5 region were removed when the reconstructions were used (Fig. 4A). This was to ensure that only the connections of the reconstructions were shown and not those of the local grafted axons, which would have biased the result. As can be seen from the figures, the connection proportions are statistically equivalent.

We then plotted the projectome of all the synthesized cells of the isocortex, without the LRAs (local axons only) in Fig. 4C and with the synthesized LRAs in Fig. 4D. The projectomes with only the outgoing connections can be seen in Supplementary Fig. S2. We can see that only by synthesizing LRAs for the MOp5 pyramidal cells has the projectome significantly changed.

In Fig. 4E, we computed the number of connections formed in the isocortex subregions by the MOp5 local axons, reconstructed and synthesized LRAs, and plotted them against the distance to each subregion. The distance to the subregions was computed as the closest distance between the border of the MOp region to the border of the other regions. We can see that, as expected, the local axons connect only to areas close to the MOp. Also, we see that the number of connections of synthesized LRAs reproduced the connections of the reconstructed LRAs well. Fig. 4F shows the distribution of out-degree of the local axons, reconstructed and synthesized LRAs. Not only the number of connections of LRAs was much higher and happened in more regions, but also the shape of the distribution was different, as can be seen in the inset of Fig. 4F. Where the local axons out-degree seemed to follow an unimodal law with small number of connections, the distribution of LRAs seemed to have a long tail towards a high number of connections. The out-degree distribution of synthesized LRAs agrees with and generalizes the out-degree distribution of the reconstructed LRAs. We found that the number of connections was proportional to the total axon lengths in brain regions; see Fig. 4G. This can also be seen in the Supplementary Fig. S4A, where the shape of the distribution of axons total lengths resembled the shape of the out-degree distribution of LRAs. This result could help predict the connectivity in brain regions. However, one should be aware that the coefficient of this relationship depends on the number of cells

synthesized in the regions, up to a limit (such as the total number of boutons of the axons or the maximum possible number of dendrites in the region).

These results show that synthesizing LRAs is essential for connecting cells locally and across distal brain regions.

**Isocortex pyramidal cells.** Finally, to showcase the possibilities of the present method, we synthesized LRAs for the pyramidal cells of all subregions of the isocortex, for which we have biological inputs to build GMM clusters. A total of 1472 biological axons were used as input for these regions: MOp ($n = 159$), MOs ($n = 393$), SSp ($n = 488$), SSs ($n = 97$), VISC ($n = 8$), VIS ($n = 83$), AUD ($n = 26$), ECT ($n = 4$), GU ($n = 11$), ORB ($n = 24$), ACA ($n = 40$), RSP ($n = 46$), FRP ($n = 7$), PL ($n = 8$), TEa ($n = 10$), PTLp ($n = 26$), AI ($n = 42$). A summarizing table showing the count per layer and hemisphere can be found in the Supplementary Material, Table S2. This amounted to a total of 21680 synthesized LRAs originating from the isocortex subregions. In Fig. 5A, B, we visualized the synthesized cells in the mouse brain. Cells are colored by the cortical layers of their somata in Fig. 5A and by region of origin in Fig. 5B. The region colors correspond to the color mapping in Fig. 5C. The projectome shown in Fig. 5C, although incomplete due to the sparsity of input data, shows a fundamentally different amount and pattern of connections than the projectome generated with only local axons in Fig. 4D. The motor and somatosensory areas showed much more efferent and afferent synaptic connections, and regions were much more interconnected overall.

## Discussion

In this work, we presented a workflow for synthesizing simulation-ready, detailed long-range axonal morphologies of the mouse brain. While we applied this method to the mouse brain, it can be adapted for rat or human brains with potentially minimal parameter tuning, depending on data availability. We demonstrated that the workflow successfully reproduced the proportions, lengths, and number of tufts in biological axons within targeted regions, with the tufts exhibiting statistically similar morphometrics. Connectivity within regions was

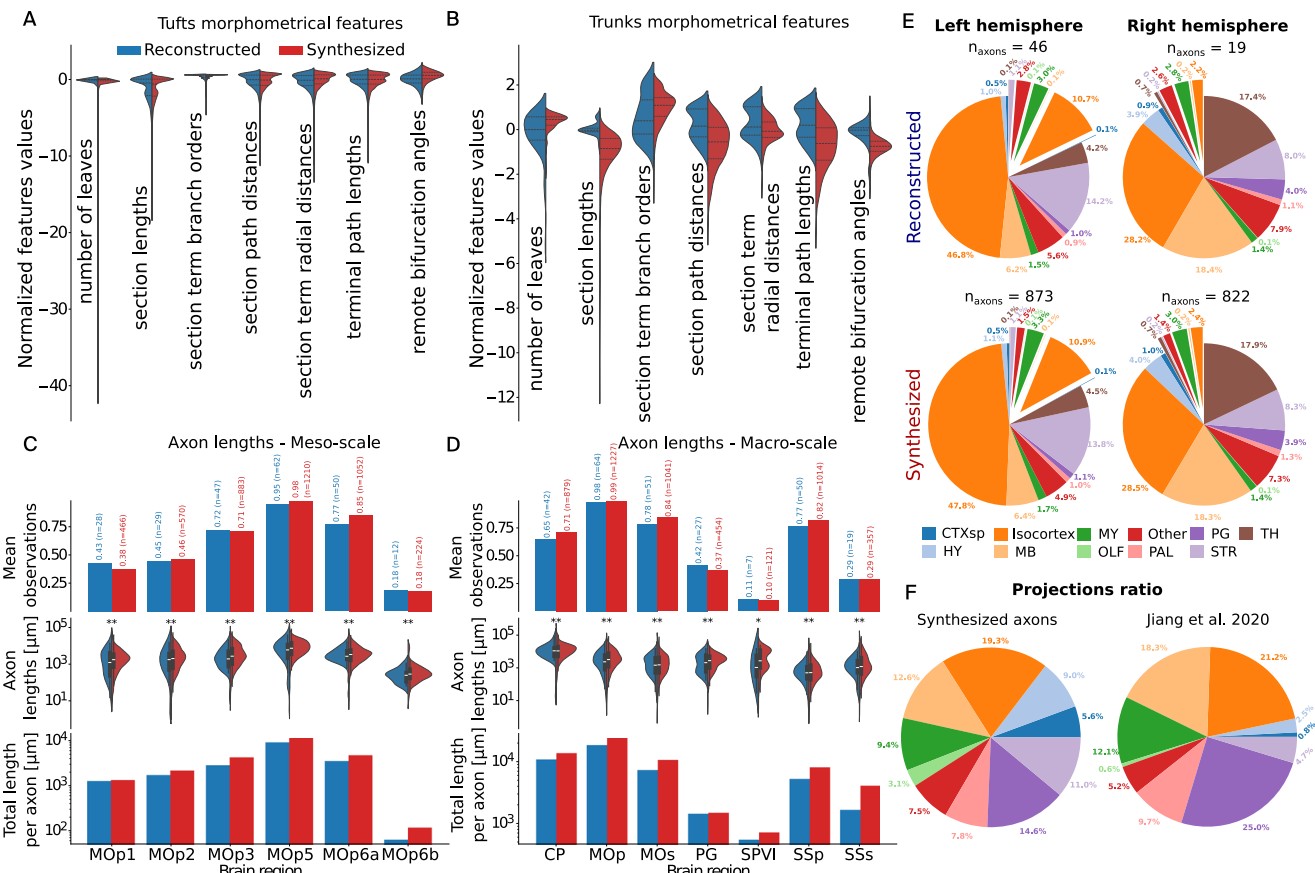

**Fig. 3 | Validation of the synthesized axons from MOp5.** Morphometrical feature distribution comparison between the tufts (**A**) and trunks (**B**) of the reconstructed (blue) and synthesized (red) axons. All distributions were centered to the reconstructions median and normalized by the reconstructions standard deviations, as done in ref. 8, see equation (1). Medians, first and third quartiles are shown (hyphens, dotted lines). **C**, **D** Comparison of axonal lengths in regions of interest between reconstructed and synthesized axons. Top: proportion of axons terminating in the regions. Middle: distribution of axon lengths. The inner boxplots represent the median (central line), the 25th and 75th percentiles (box edges), and the whiskers extend to the most extreme data points within 1.5× the interquartile range (IQR) from the box. Bottom: total length normalized by the number of axons.

**C** focuses on the MOp layers and D on distal regions. The boxes in the boxplot show first and third quartile, and median value. The stars show strong significance (**) and non-randomness (*) when the Maximum Visible Spread (MVS) score between the populations is <0.1, respectively <0.5 **E** Comparison of tufts locations clustered from the reconstructed (top) and synthesized (bottom) MOp5 axons. The exploded parts in the pie charts represent projections in contralateral regions. **F** Projections ratio comparison between our synthesized axons and reconstructed MOp5 axons analyzed in ref. 27. CTXsp Cortical subplate, PAL Pallidum, HY Hypothalamus, TH Thalamus, MB Midbrain, MY Medulla, OLF Olfactory areas, PG Pontine gray, STR Striatum.

accurately replicated, as shown in ref. 8, where reproducing similar dendritic morphologies led to equivalent local connectivity between biological and synthesized circuits. Long-scale connectivity is also consistent with experimental data. In our validation instance of the MOp5 region, axons targeting was even consistent with morphologies not included in the algorithm inputs. Our algorithm is thus capable of generating full brain connectivity predictions, that can be compared against and completed from future experiments, once more data becomes available.

Small differences in axon lengths and resulting connections are likely due to the random placement of tuft target points within the target regions, occasionally causing tufts to extend beyond the intended areas. This issue could be mitigated by either constraining tuft growth within the target regions or positioning them closer to their common ancestor in the reconstructed dataset. In addition, as discussed in ref. 23, improvements in the synthesis algorithm will improve the agreement between synthesized and biological axons. These include a refined version of the Steiner tree algorithm to incorporate more constraints, as well as an optimization process to select the most appropriate input parameters based on the input datasets. In addition, as more long-range axonal data become available, we will be able to better approximate biological rules,

and improve the ability of our algorithm to generalize to new inputs.

Our approach is versatile, due to the small number of parameters, that require minimum tuning from the user. By focusing on replicating the biological data while allowing for statistical variability, we enable properties such as connectivity to emerge naturally, rather than imposing them onto the model. However, this dependence on biological data also introduces some limitations. We are constrained by the projection patterns present in the input dataset, thus we can only accurately synthesize axons for the entire brain if all relevant projection patterns are represented. Additionally, the targeting of the synthesized axons is sensitive to the quality of the reconstructed axons in the dataset. In the future, we will expand this methodology to generalize the observed patterns in the full brain, also linking the observed projection patterns to gene expression through transcriptomic datasets[31–33]. In addition, the clustering method could be improved by considering region hierarchy or spatial location within regions in the clustering step. Finally, we could leverage the Gaussian nature of the clusters to sample the lengths in the synthesis step.

One of the key advantages of our algorithm is its ability to synthesize whole-brain axons, leading to a comprehensive model of full-brain connectivity. Therefore, we enable the computational synthesis

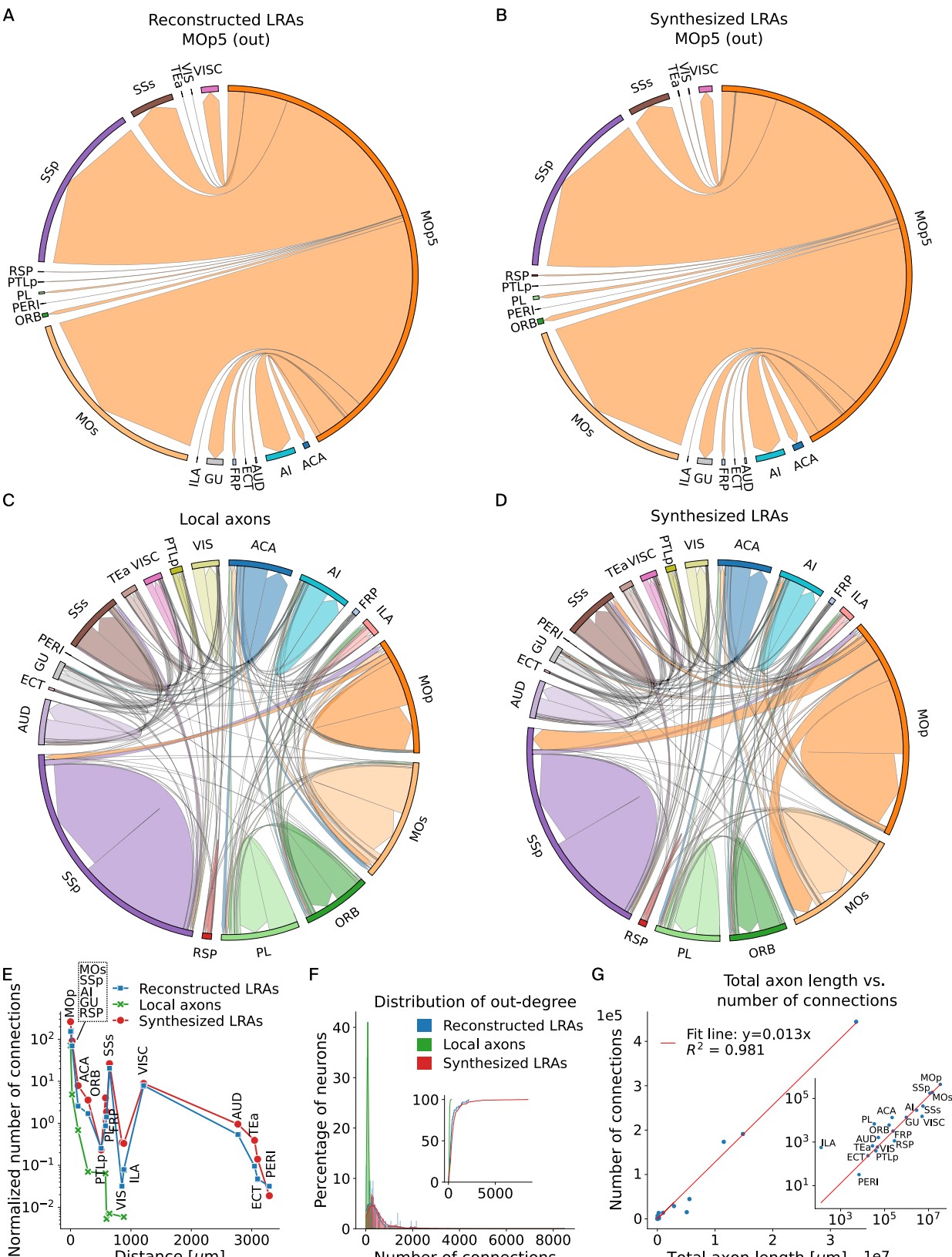

**Fig. 4 | Connectivity of the synthesized MOp5 axons.** Proportions of outgoing connections from MOp5 pyramidal cells, using the 65 biological reconstructions (**A**) and the synthesized LRAs (**B**). Projectome of the isocortex subregions, with local axons only (**C**), respectively, with long-range axons for the MOp5 pyramidal cells (**D**). **E** Number of connections in subregions of the isocortex versus distance to subregion, for biological LRAs (blue), local axons (green) and LRAs synthesized with the present method (red). As can be expected, local axons connect only to regions close to the MOp. **F** Out-degree distribution of the biological LRAs (blue), local axons (green), and synthesized LRAs (red). Inset: cumulative histogram. **G** Total number of connections vs. length of the synthesized long-range axons in the subregions of the isocortex. A linear relationship seemed to describe the observed data well. LRAs long-range axons.

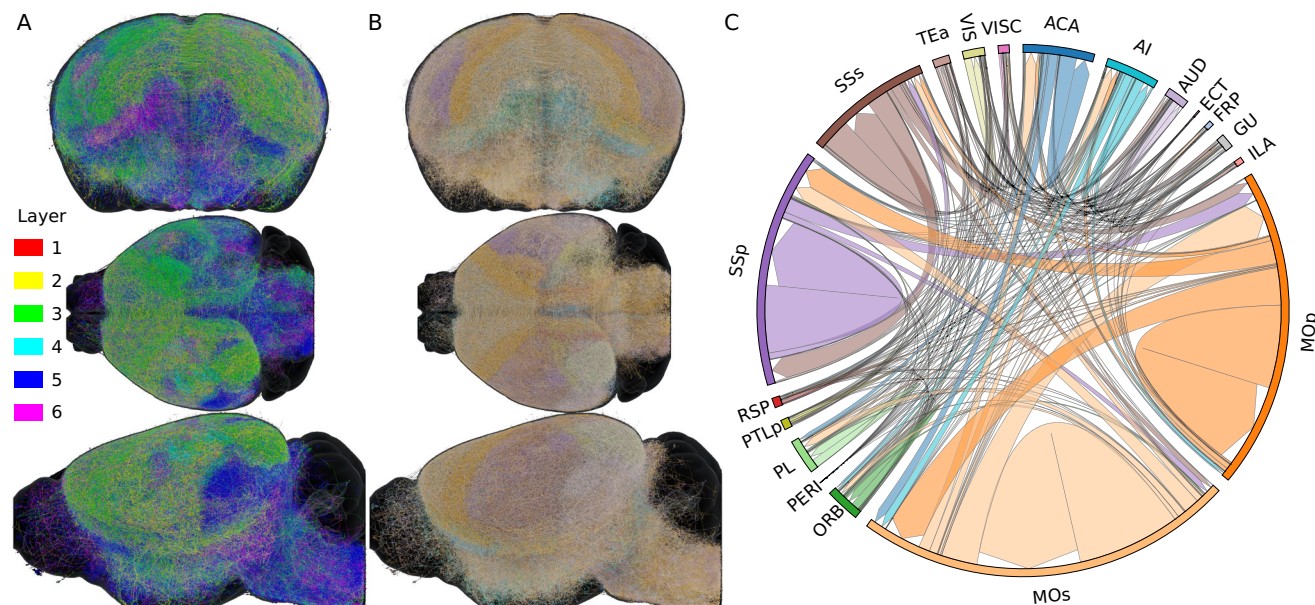

**Fig. 5 | Synthesis of 21680 axons for pyramidal cells in subregions of the iso-cortex.** The list of regions for which we had biological data to synthesize the axons can be found in the Supplementary Table S2. **A**, **B** Visualizations in the mouse brain of the synthesized axons, colored by layer, respectively by region, of their somata. 25% of the synthesized axons are shown. The color code of (**B**) is the same as (**C**). The tuft synthesis not being strictly constrained to the brain volume, a few tufts can be seen growing out of it. **C** Projectome of the long-range axons. Arrows show the direction of connection (from pre- to post-synaptic), and their size is proportional to the number of connections.

of a full brain, once a more complete axonal projection dataset becomes available. By integrating additional biological data, we will refine and extend our models to simulate more complex and realistic brain-wide connectivity. The scalability and adaptability of our method make it a valuable tool for future research, bridging the gap between current experimental limitations and the goal of creating a fully synthetic, computational model of the brain for computational simulations[2,34].

Looking further into the future, our methodology holds significant promise for advancing healthcare applications. The generation of highly accurate and biologically informed neural circuits can significantly enhance patient diagnostics, particularly for conditions involving abnormal brain connectivity. An interesting future direction is to incorporate genetic information[35] in the clustering algorithm for the definition of axon targeting and the generalization to axon types for which morphological reconstructions are lacking. In addition, our algorithm will be useful for large-scale MRI simulations[36] to improve neuro-imaging predictions for improving patient diagnostics. This approach could allow for earlier detection of neurological disorders by simulating disease progression and identifying biomarkers before symptoms manifest. Additionally, the method's scalability and ability to synthesize diverse axonal projections can contribute to more efficient drug discovery by facilitating in silico testing of therapeutic compounds across realistic brain networks. Furthermore, the model's adaptability to individual datasets makes it well-suited for personalized medicine by facilitating the simulation of patient-specific brain connectivity and drug responses, which could revolutionize treatment strategies and optimize long-term outcomes for patients.

## Methods
### Nomenclature
In this work, we used the terms "brain region" in a broad sense: it might label entire regions (thalamus, cerebellum), sub-regions (primary, secondary motor areas), or layers separation (primary motor area, layer 5, layer 6).

We used brain regions acronyms defined in the Allen Brain Atlas[37]. We spelled out each acronym that is helpful for this work's comprehension when they were used. We also provided an exhaustive list of all brain acronyms used in the Supplementary section 10.1.

### Long-range axon synthesis
We presented an algorithm for synthesizing LRAs in ref. 23. This method was able to synthesize cells accurately mimicking reconstructed axonal morphologies. We review here the main relevant steps of this algorithm; see Fig. 6. Reconstructed axonal morphologies are taken as input. Terminations that are close together (within a radial and path distance) are clustered up to their common ancestors in so-called *tufts*. The axons with all tufts taken out are labeled as *trunks*. The somata of the reconstructed morphologies are used as *source points*, and the locations of the tufts ancestors as *target points*. A graph is created with these and additional points taken as vertices (see ref. 23 for details). Then, we run the weighted Steiner tree algorithm[38] on this graph, which connects the targets while minimizing cable length, and allowing to prefer edges, such as ones that would be inside fiber tracts. The created trunk is then post-processed to reproduce local morphometrics of biological axons, with operations such as adding random noise and taking into account local history for modifying the curvature. Finally, the tufts that were initially clustered are synthesized based on their topological properties, with the method previously described and validated for synthesizing dendritic morphologies[8].

We could assess that this algorithm was able to produce morphometrical properties that matched with the reconstructed axons at the trunk, tuft, and morphology levels, see ref. 23.

### Axonal projections analysis
In our previous work[23], the source points and the target points were taken directly as the somata and common ancestors of tufts of the reconstructed morphologies, respectively. Furthermore, the topological signature of the tufts of the biological axon to mimic was used for their synthesized analogs. In the present work, we generalized the selection of source points, target points, and tufts from populations of biological axons rather than from individual morphologies. This eventually allowed us to synthesize and connect neurons in the entire mouse brain, within and between brain regions.

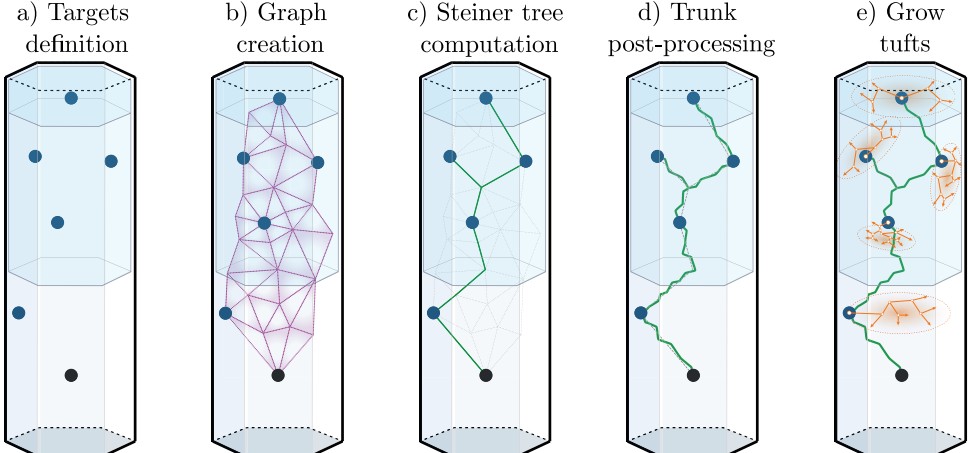

**Fig. 6 | Main steps of the long-range axon synthesis.** From an input biological axon, we take the soma as the source point (**a**), cluster the tufts, and make their common ancestors the target points. The synthetic trunk is created by connecting the targets from the source point with the Steiner tree on a graph method (**b**, **c**). The trunk is then post-processed (**d**) with the addition of noise to reproduce local morphometrics of biological axons. Finally, the tufts are grown (**e**) at the target points from their topological properties, with the same method as dendrites[8].

We explain our methodology in what follows, see Fig. 7 for a visual schematic.

### Input data

**Morphologies.** We used morphologies with long-range axons from three different sources:

- 1084 morphologies from the Janelia MouseLight Project[25].
- 1741 morphologies from ref. 24.
- 800 unpublished morphologies from collaboration with Southeast University, Nanjing, China and H. Peng (using the same protocol as ref. 24).

For the latter set unpublished morphologies, the protocol used was the same as ref. 24. In other words, we used transgenic mice that contain a combination of the following individual driver and reporter lines: Cux2-CreERT2, Fezf2-CreER, Gnb4-IRES2-CreERT2, Plxnd1-CreER, Pvalb-T2A-CreERT2, Tnnt1-IRES2-CreERT2, Vipr2-IRES2-Cre-neo, Snap25-IRES2-Cre, Slc17a7IRES2-Cre, Esr2-IRES2-Cre, Ai139, Ai140, Ai82, Ai166, Ai14, Ai65F, and RCL-Sun1sfGFP. All transgenic mice were maintained in C57BL/6J congenic background. For each genotype of transgenic mice, we used both male and female mice, ages ranging from 8 weeks to 5 months old. Mice were housed in animal rooms on a 14/10 h light/dark cycle (6 am–8 pm light). The room temperature was set at 70 °F (21 °C) and the relative humidity at 40%. The study did not involve wild animals. All experimental procedures using live animals were performed according to protocols approved by Institutional Animal Care and Use Committee (IACUC) of the Allen Institute for Brain Science.

Morphologies could be incompatible with our methodology for reasons such as incomplete reconstructions or artifacts due to the reconstruction techniques. Therefore, we post-processed them with the `Repair` workflow of `Morphology-Workflows` software[39], which applied corrections such as repairing out-of-plane cut branches or removing unifurcations. 16 (8 from ref. 25 + 8 from ref. 24) morphologies did not pass all the correcting processes, and 8 (1 + 5 + 2) did not pass the axonal projection analysis (due to being detected out of bounds or having faulty axons). In total, 3601 morphologies were used in this work.

**Mouse brain atlas.** We used the mouse brain atlas described in ref. 29, which is an enhancement of the Common Coordinate Framework version 3 (CCFv3) atlas of the Allen Brain Institute[37]. Notable additions are the annotations of the barrel field areas and the distinction of cortical layers 2 and 3.

**Projections computation.** In order to generalize the axon synthesis algorithm, we first computed the projections of reconstructed biological axons by counting the number of terminals of each axon in all regions they terminated and computing the axon path lengths in these regions. Our method then uses one of these features to cluster the axons with somata in the same source region.

Let us formalize this problem in the following way. We want to classify a set of $N$ biological neurons. Let $a$ be the axon of a neuron $n$. We denote $s_a$ as the source brain region where the soma of neuron $n$ is located. The axon $a$ projects and terminates in a number of target brain regions, $t_a \in (\mathbb{N})^B$ is the vector counting the number of terminal points in all brain regions $B$. In other words, $t_a^{(b)}$ is the number of terminal points of axon $a$ into brain region $b$. We define in a similar way $l_a \in (\mathbb{R}^{0+})^B$ the path length of axon $a$ inside brain region $b$. Note that brain regions can be defined at various levels of detail according to the hierarchy of the brain atlas. Let us further denote $f_a$, the feature vector for the classification of neuron $a$. We consider the case where $f_a = l_a$. $s_a$ is not included in $f_a$ because we impose a separate classification for each source region.

**Clustering method.** There are several methods for unsupervised clustering of data such as K-means and its variants (K-medoids, K-centroids[40]) and hierarchical clustering based on statistical difference of the subclasses[28]. We chose here to assume that our data could be described by normal probability density functions, using GMMs[41]. Let us imagine that region $s_a$ has $C \in \mathbb{N}$ clusters. To assume that $a$ originates from a GMM is equivalent to saying that the probability of $a$ to belong to cluster $c$ is given by :

$$P(c|a) = \frac{P(c)P(a|c)}{P(a)} \tag{2}$$

$$= \frac{p_c \mathcal{N}(f_a, \mu_c, \Sigma_c)}{\sum_{k=1}^{C} p_k \mathcal{N}(f_a, \mu_k, \Sigma_k)}, \tag{3}$$

where $\mathcal{N}(f_a, \mu_c, \Sigma_c)$ is the multivariate normal distribution with mean $\mu_c$ and covariance matrix $\Sigma_c$:

$$\mathcal{N}(t_a, \mu_c, \Sigma_c) = \frac{1}{\sqrt{(2\pi)^B \det(\Sigma_c)}} \exp\left(-\frac{1}{2}(f_a - \mu_c)^T \Sigma_c^{-1}(f_a - \mu_c)\right). \tag{4}$$

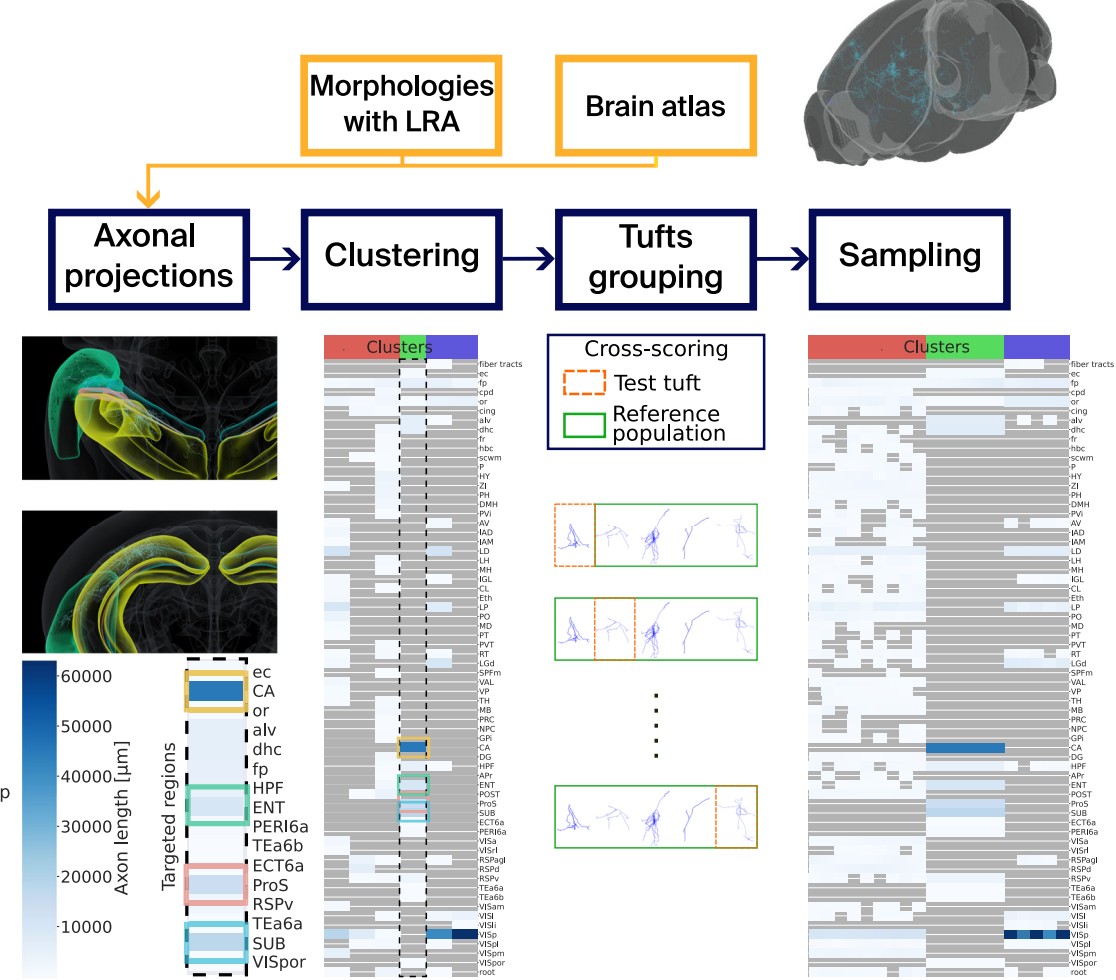

**Fig. 7 | Schematic view of the axonal projections analysis.** Neuronal morphologies with long-range axons (LRAs) placed in a mouse brain atlas are taken as input. The projections of the LRAs are computed, and we cluster the LRAs based on these projections with GMMs. Tufts are separated from the trunks of the LRAs, and they are counted and assigned a score for each cluster and target brain region. Finally, we can use the GMMs to sample target axon lengths for every target brain region defined in the clusters. fp: corpus callosum, posterior forceps.

We assume $\Sigma_c$ symmetric and positive definite $\forall c \in C$. Note that we have the constraint $\sum_{k=1}^{C} p_k = 1$.

A common technique[42] to optimize the clustering is to maximize the likelihood, or log-likelihood, of the observed data, based on the parameters of clustering $\theta$[42]. Here, we can write $\theta = \{p_1, \ldots, p_C, \mu_1, \ldots, \mu_C, \Sigma_1, \ldots, \Sigma_C\}$. The log-likelihood of the data based on the parameters of clustering is given by (assuming $a$ are iid) :

$$l(\theta) = \log\left(\prod_{a=1}^{N} P(a)\right) = \log\left(\prod_{a=1}^{N}\sum_{k=1}^{C} p_k \mathcal{N}(f_a, \mu_k, \Sigma_k)\right) := \log\left(\prod_{a=1}^{N} P(f_a|\theta)\right). \quad (5)$$

However, optimizing the log-likelihood (5) is not tractable in the case of large GMMs. Instead, we used the Expectation-Maximization (EM), see Supplementary Material 10.2.1.

The number of clusters $C$ for a source region can be either imposed or selected to maximize a score, the BIC, see Supplementary Material 10.2.2. Since we were less interested in describing the biological data in this work than reproducing it, the number of clusters $C$ per source region $s$ is optimized on the BIC score within a range of $C$ going from the number of axons $N_s$ in $s$ divided by 2, to $N_s$, unless specified otherwise.

**Tufts grouping.** Once the GMM clusters were defined for all source regions of the input axons, the tufts were clustered using the previously described clustering algorithm, section "Long-range axon synthesis" and ref. 23. We used a maximum clustering radius and path distance of 300 µm. The tufts were then grouped by GMM cluster and region of their common ancestors. For each group $g$, we computed the average $\bar{N}_{\text{tufts}}^{(g)}$ and variance $\sigma^2(N_{\text{tufts}}^{(g)})$ of tuft numbers for each group. Finally, the tufts were assigned a *representativity score* within their group, which is a measure of how close they are to the other tufts in their group in terms of a set of morphometrical features. In this work, all morphometrics were computed using `NeuroM`[43]. We used the MVS score to measure the similarity of the features. The calculation details can be found in Supplementary Material 10.3.

**Sampling.** Finally, one can draw samples from the GMMs to verify the clustering. To do so, we first chose one distribution (or cluster $c$) from the mixture, our choice weighted by the probability $p_c$. It is then possible to generate the vector of lengths $l_a$ or terminal points $t_a$ by drawing a sample from the chosen distribution:

$$l_a \sim \mathcal{N}(\mu_c, \Sigma_c). \quad (6)$$

We added a post-processing step to all samples, which removes values sampled in regions not observed in the biological input data.

We used the sampling here as an indicator of the clustering accuracy only. However, one could use the sampling to synthesize

directly axons with the sampled lengths, for instance by choosing tufts to add up to all lengths in each targeted region.

### Synthesizing in the mouse brain

We now present how we used the axonal projections analysis presented in section "Axonal projections analysis" as input data for the synthesis algorithm in section "Long-range axon synthesis" to synthesize LRAs in the mouse brain.

**Initial morphologies synthesis.** First of all, we synthesized neuronal morphologies made of somata, dendrites, and grafted reconstructed *local* axons in the isocortex, with the same methodology as in refs. 8,44. These local axons were copied from previous experimental reconstructions and grafted to the synthesized cells. Since mostly pyramidal cells of the mouse cortex project to distal regions[45], we filtered pyramidal cells of the brain region $s_a$ and would replace their local axon with a synthesized LRA. We synthesized LRAs for all pyramidal cells in the regions for which we synthesize, but that can be changed to a provided portion of them.

**Source points.** The somata of the filtered pyramidal cells were used as the source points of the synthesis algorithm. We assigned a GMM cluster for each source point by randomly picking a cluster $c$ with probability $p_c$ from the clusters $C$ of source $s_a$.

**Target points.** We computed the probability of an axon to target a brain region $b$ as the number of axons targeting $b$, divided by the total number of axons in the cluster. For a picked target brain region, a number $T_a^{(b)} \sim \mathcal{N}\left(\bar{N}_{\text{tufts}}^{(g)}, \sigma(N_{\text{tufts}}^{(g)})\right)$ of target points were randomly placed inside $b$. All the targets of axon $a$ were then connected with a better edge weight for targets inside fiber tracts to form the trunk. In this way, the trunk would preferentially follow the fiber tracts.

**Tufts selection.** Finally, tufts were selected for each target point with probability computed based on the representativity scores of their group $g$, and then synthesized with the `NeuroTS` software[8].

**Creating connections.** Once all the morphologies were fully synthesized, we made axo-dendritic connections as described in ref. 2. In a few words, touches were detected based on the physical proximity of neurite branches and filtered based on a minimum inter-bouton interval. Touches were then pruned based on physiological synapse density and converted into synapses. Connections and connectivity matrices were analyzed using the `ConnectomeUtilities` software[46].

### Reporting summary

Further information on research design is available in the Nature Portfolio Reporting Summary linked to this article.

## Data availability

The data used and generated in this study have been deposited in the Zenodo database https://doi.org/10.5281/zenodo.13790069.

## Code availability

The code for clustering the axons can be found at https://github.com/BlueBrain/axon-projection. The scripts for data analysis and plotting the figures is available at https://github.com/Remy2506/axon_projection_figures.git.

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

## Acknowledgements

This study was supported by funding to the Blue Brain Project, a research center of the École polytechnique fédérale de Lausanne (EPFL), from the Swiss government's ETH Board of the Swiss Federal Institutes of Technology. We further thank Cyrille Favreau, Fabien Petitjean, and the Scientific Visualization team for developing the visualization tools that helped produce the rendering of circuits within the mouse brain, and the Data Knowledge Engineering team for helping manage the morphologies datasets.

## Author contributions

R.P., A.B., L.K., and H.M. contributed to the algorithm conception and design. Data collection was performed by R.P. and H.P. R.P. performed the data analysis. H.M. secured funding and supervised the research project. R.P. and L.K. wrote and revised the manuscript. All authors read and approved the final version of the article.

## Competing interests

The authors declare no competing interests.
