## [Transparent Peer Review file · Nature Communications]

Generating brain-wide connectome using synthetic axonal morphologies

Corresponding Author: Dr Remy Petkantchin

Version 0:

Reviewer comments:

Reviewer #1

(Remarks to the Author)

The authors present an ambitious method for determining local neural connectivity across multiple spatial scales by synthesising the morphologies of long range axons. They demonstrate their method by comparing the morphometrics and modelled connectivities of their synthetic cells with those of a recent dataset of reconstructed axons. This study is an important step towards modelling detailed connectivity between different brain regions, but some of the statements in the current text could be seen as overoptimistic given the current results and data availability. The methodology used in the paper is sound, but future work should address the sensitivity of the desired activity simulations to the features that might differ between the synthesised and reconstructed cells, especially the reduced heterogeneity in morphometrics and increased numbers of postsynaptic partners in some modelled cells.

Otherwise, I have only minor comments on the study.

- Line 19: 'predict whole-brain connectivity at a single cell level'. I think the current study is very far from this as the model and data only apply to certain subsets of axons and it is unclear how well the connectivity of the cells it does model is reproduced given the complexity of the model and lack of available data.
- Line 41: 'However, the majority... brain topology'. I find this sentence very unclear. The authors should be more specific on what they mean by 'brain topology'.
- Line 61: 'progressing from the connectome... to the projectome'. This sentence seems to imply that inter-areal connectivity is poorly understood, when techniques such as DTI and fMRI provide plenty of data on this. The authors should be more specific that they mean long-range connectivity at a cellular level.
- Line 84: 'proportions of targeting against experimental data': Unclear what is meant.
- Fig 1D: 'A linear relationship'. This is hard to check as the data is plotted on a log-log plot. There appears to be a non-linear power-law relationship.
- Fig 2B: Some of the synthetic axons leave the brain entirely. The authors should address this in the text.
- Fig 3A, B : There appears to be a general trend that synthetic axons are less variable than the data. Could the authors comment on this?
- Fig 4F. The shaded region in the inset histogram is unnecessary and makes the results harder to see. Further, the synthesised cells have a heavier tail (towards high numbers of outgoing connections) than the reconstructed cells. Would this have a significant effect on simulated activity using synthetic axons?
- Fig 5. Axons leave the brain again.
- Line 247: 'adapted... minimal parameter tuning'. I do not believe that this statement is justified by the results shown.
- Line 338: 'The formalism...': This is a very unclear sentence.

(Remarks on code availability)

The code is freely available and well-documented.

Reviewer #2

(Remarks to the Author)

This study nearly identical to one recently published by the same group. Many of the figures and diagrams and many whole

sentences in this manuscript are in the published study

<https://link.springer.com/article/10.1007/s12021-024-09696-0>

Thus, the paper under review should not be published as the majority of it has already been published

(Remarks on code availability)

This study nearly identical to one recently published by the same group. Many of the figures and diagrams and many whole sentences in this manuscript are in the published study

<https://link.springer.com/article/10.1007/s12021-024-09696-0>

Thus, the paper under review should not be published as the majority of it has already been published

Reviewer #3

(Remarks to the Author)

In this manuscript, Petkantchin and colleagues detail the development and application of algorithms and a workflow to use experimentally derived morphological and connectomic datasets for generating representative, large-scale synthetic morphologies of local (inter-regional) and long-range (brain-wide) axonal structures. Using published datasets of validated, brain-wide neuronal reconstructions (dendrites and axons) from Janelia's MouseLight Project (1,084 morphologies) and the Allen Institute for Brain Science's fMOST data (1,741 morphologies), as well as 800 unpublished morphologies provided by Peng, one of the co-authors, the authors first applied an unsupervised Gaussian Mixture Model (GMM) to cluster the axonal projections based on their regions of origin and various regions of termination. The clustering results were used to generate synthetic projections, onto which axonal tufts (terminal arbors) for both local and long-range projections were synthesized using a previously described algorithm that was first developed to generate synthetic dendrites. To validate the projection clustering, modeled connectivity, and algorithm for synthesizing axonal terminal morphologies, the authors provide metrics of the connectivity and morphometrics of 1,695 synthetic pyramidal neurons originating from Layer 5 of the primary motor cortex (MOp5), compared against experimentally reconstructed examples from the aforementioned reconstruction datasets. Additionally, the authors synthesized 1,472 long-range axons based on GMM clusters from other major cortical subregions to compare the accuracy of the modeled connectivity relative to the experimental datasets. In both applications, quantitative comparison of key metrics of both the synthesized connectivity and tuft morphologies showed highly accurate, statistically significant reproduction of the experimentally derived axonal data from the targeted cortical regions.

In summary, the study is thorough, offers a significant contribution, and introduces a highly innovative approach to addressing a complex challenge within the neuroscience community, i.e. creating more complete single-neuron axonal morphologies. This method still faces limitations due to the current availability of fully reconstructed neurons (both dendrites and axons). However, as additional experimental datasets with more training neurons become available, this model/framework has the potential to be highly effective for synthesizing neurons and their projection targets. Thus, with minor revisions, this paper should be a good candidate for publication in Nature Communications.

Minor issues.

- It would be helpful if Figure 1 included a schematic representation of the pipeline, outlining the key steps in developing the algorithm for synthesizing single neurons, the main stages of the automated synthesis process. They should also provide examples of synthesized outputs compared to real examples to illustrate the accuracy of their algorithm.
- One of the major components of this work is the synthesis of axonal terminal arbors using an algorithm originally developed to model dendritic processes. Given that axonal terminals are typically much more complex and denser than dendrites, the authors should provide a detailed explanation of their approach and present data to demonstrate the accuracy of the synthesized axonal terminals. This would help validate the robustness of their algorithm for modeling such intricate structures.
- The authors should clarify whether their algorithm is exclusively suited for the synthesis of cortical pyramidal neurons or whether it can be applied to other neuron types in the brain. If the latter is true, it would be useful to provide examples validating the algorithm's applicability to other brain cell types (e.g., neurons in the striatum, thalamus, or cerebellum).
- Similarly, the authors note that their proposed method for synthesizing axons is species-agnostic, suggesting it can be applied to both rats and humans with some parameter tuning. However, it is unclear whether they have demonstrated this approach in species beyond mice. Including examples of the algorithm's application to other species would strengthen their claim.
- Tuft grouping was performed using a maximum clustering radius and path distance of 300 microns. Did the authors test different threshold values before selecting these parameters?

(Remarks on code availability)

Version 1:

Reviewer comments:

Reviewer #1

(Remarks to the Author)

I believe that the authors have generally responded well to my own comments. I also find novelty in the application of their model to long-range connectivity and in its comparison to a relatively large dataset.

(Remarks on code availability)

Reviewer #3

(Remarks to the Author)

The authors have fully addressed all the issues I raised previously. The manuscript is appropriate for publication in Nature Communications.

(Remarks on code availability)

Response to reviewers' comments:

Reviewer #1 (Remarks to the Author):

The authors present an ambitious method for determining local neural connectivity across multiple spatial scales by synthesising the morphologies of long range axons. They demonstrate their method by comparing the morphometrics and modelled connectivities of their synthetic cells with those of a recent dataset of reconstructed axons. This study is an important step towards modelling detailed connectivity between different brain regions, but some of the statements in the current text could be seen as overoptimistic given the current results and data availability. The methodology used in the paper is sound, but future work should address the sensitivity of the desired activity simulations to the features that might differ between the synthesised and reconstructed cells, especially the reduced heterogeneity in morphometrics and increased numbers of postsynaptic partners in some modelled cells.

Otherwise, I have only minor comments on the study.

• Line 19: 'predict whole-brain connectivity at a single cell level'. I think the current study is very far from this as the model and data only apply to certain subsets of axons and it is unclear how well the connectivity of the cells it does model is reproduced given the complexity of the model and lack of available data.

This sentence was modified to narrow the resulting conclusion. However, we think the model still proposes a prediction, which in time can be proven correct or wrong.

• Line 41: 'However, the majority... brain topology'. I find this sentence very unclear. The authors should be more specific on what they mean by 'brain topology'.

We thank the reviewer for his comment. We corrected it using more accurate language. By "brain topology", we meant the geometry of the brain, its bounding volume, and forbidden or preferential areas.

• Line 61: 'progressing from the connectome... to the projectome'. This sentence seems to imply that inter-areal connectivity is poorly understood, when techniques such as DTI and fMRI provide plenty of data on this. The authors should be more specific that they mean long-range connectivity at a cellular level.

We thank the reviewer for pointing out this perspective. In fact, we wanted to stress out that our approach was "bottom-up", as we work at the cellular level. We modified the sentence accordingly.

• Line 84: 'proportions of targeting against experimental data': Unclear what is meant.

We thank the reviewer for pointing out this ambiguity. We expect this is clearer with our modification.

• Fig 1D: 'A linear relationship'. This is hard to check as the data is plotted on a log-log plot. There appears to be a non-linear power-law relationship.

The graph is indeed on a log-log scale, however the fitted relationship is a proportional law, and the coefficient is very close to 1, with $R^2 = 0.996$. The proportional relationship also follows our intuition, that the more terminals there is in a region, the greater cable length in the region. This result is not central to the article, and might present some controversy, as the reviewer noted.

• *Fig 2B: Some of the synthetic axons leave the brain entirely. The authors should address this in the text.*

We believe the reviewer refers to biological axons and not synthetic. Indeed this is a point we did not clarify. The biological reconstructions might have some placement errors, due to the reconstruction method used, and the Atlas space in which the axon is registered which might vary between subjects. A sentence was added in the legend to address this issue.

• *Fig 3A, B : There appears to be a general trend that synthetic axons are less variable than the data. Could the authors comment on this?*

There appears to be less variation indeed in the trunks morphometrical features for the synthesized cells. This is related to the method of generating the trunk which is identical for cells, whereas the morphometrical features of biological cells might vary per specimen (different mice were used) or per cell spatial location. In our approach, we define the location only by the broader brain region the cell is in, and place synthetic cells within this volume. Furthermore, at this stage of our method, we did not fit the parameters of the trunk generation algorithm to the best match given data. This is because this optimization might take huge computing time, and the amount of biological data does not justify it just yet.

• *Fig 4F. The shaded region in the inset histogram is unnecessary and makes the results harder to see. Further, the synthesised cells have a heavier tail (towards high numbers of outgoing connections) than the reconstructed cells. Would this have a significant effect on simulated activity using synthetic axons?*

We removed the shaded area in the inset of the figure, it is indeed clearer now. It is correct that it seems that the synthesized cells have a longer tail, we did not investigate its cause. It might seem this way maybe because of the lower number of reconstructed morphologies. As we can see in Fig. 4F, the distribution for the reconstructed morphologies is more sparse compared to the one for synthesized morphologies. If there is indeed a longer tail, in order to estimate its effect on electrical activity, we would need to first understand if the axons create more connections uniformly to every targeted region and cell types, or heterogeneously. In the latter case, it could be possible that some synthesized axons create connections where they should not. Looking at Fig 4E, we can suspect that these extra connections are made to regions such as VIS, TEa, ECT (and ACA, ORB to a lesser extent). However, looking at the normalized number of connections to each region in Fig 4B, we can see that this asymmetric effect seems minimal. Thus what we hypothesize is that there might potentially be extra-stimulation by the synthetic axons in some regions, but that this effect should not be statistically significant.

• *Fig 5. Axons leave the brain again.*

In the synthetic case, this might happen when tufts root points (“common ancestors”) are placed too close to the brain boundary because tufts are not strictly constrained inside it. There is no easy solution to that problem. Potential solutions are:

- Forcing the tufts to stay inside the brain, no matter what. Problem: the shape of the tufts could drastically change, thus morphometrical features of the tufts would not match the biological data, and connectivity would be affected;
- Choosing only tufts that “fit in” the volume, given a root. Problem: this would bias the chosen tufts, and thus also the morphometrical features and connectivity;
- Placing tufts roots further from the boundary. Problem: Would affect connectivity, and depending on tuft orientation: how far from the boundary?

Discussing this issue in the present manuscript is slightly beyond the scope of this work, and the implemented solution is described in the methodology paper Ref. 25 (section “Boundary Constraints”). We added a comment on that in the legend of Fig 5.

• Line 247: ‘adapted... minimal parameter tuning’. I do not believe that this statement is justified by the results shown.

We thank the reviewer for that remark. Indeed we cannot prove that it would indeed work “out-of-the-box” for other species. However in the whole algorithmic pipeline that we implemented, there is no explicit fitting to the mouse species. The only requirement would be to have data available in the same format, i.e. morphologies and a brain annotation space. It might be needed to adjust the input parameters if one would want to fit morphometrical features on the input data. However, this is out of scope for this study, due to lack of sufficient experimental data. We have modified this statement to clarify this point further in the manuscript.

• Line 338: ‘The formalism...’: This is a very unclear sentence.

We thank the reviewer for pointing out this issue. We removed this sentence because it did not bring any additional information.

Reviewer #1 (Remarks on code availability):

The code is freely available and well-documented.

Reviewer #2 (Remarks to the Author):

This study nearly identical to one recently published by the same group. Many of the figures and diagrams and many whole sentences in this manuscript are in the published study

<https://link.springer.com/article/10.1007/s12021-024-09696-0>

Thus, the paper under review should not be published as the majority of it has already been published

Reviewer #2 (Remarks on code availability):

This study nearly identical to one recently published by the same group. Many of the figures and diagrams and many whole sentences in this manuscript are in the published study

<https://link.springer.com/article/10.1007/s12021-024-09696-0>

Thus, the paper under review should not be published as the majority of it has already been published

We thank the reviewer for pointing out that the novelty of the manuscript was not clear. We have significantly modified the introduction to make the following points of novelty more apparent.

The work Ref 25 gives a detailed description of the single-axon synthesis algorithm. Our previous paper focused on the computational and algorithmic aspects of the algorithm and lacks a concrete biological application and validation. The results and discussion of the two studies take completely different directions.

We stress that in both works we need an algorithm for axonal synthesis across the whole brain. However, the similarity of the two papers ends at the long-range axon synthesis algorithm.

In this paper, we resolve the problem of formalizing axon targeting, to reproduce the inter-regional connectivity. The present work brings among others the following novelties:

- New experimental data (800 morphologies) focusing on specific regions of interest, which are essential for modeling of the cortical axons and their validation.
- The methodology for clustering input axons of the same source region, using Gaussian Mixture clustering. This allows for selecting “target points” of the synthesis algorithm according to the biological data. Target points were selected arbitrarily in the previous work.
- The probability of selection of the barcode of tufts to synthesize, based on the representativity of the tufts’ metrics inside their Gaussian cluster.
- The creation and validation of connectivity metrics, per axon and per region.

Therefore, we complete the pipeline to go from single cells to full brain connectome, which brings the novelty of having full brain connectivity at single cell resolution.

Reviewer #3 (Remarks to the Author):

In this manuscript, Petkantchin and colleagues detail the development and application of algorithms and a workflow to use experimentally derived morphological and connectomic datasets for generating representative, large-scale synthetic morphologies of local (inter-regional) and long-range (brain-wide) axonal structures. Using published datasets of validated, brain-wide neuronal reconstructions (dendrites and axons) from Janelia’s MouseLight Project (1,084 morphologies) and the Allen Institute for Brain Science’s fMOST data (1,741 morphologies), as well as 800 unpublished morphologies provided by Peng, one of the co-authors, the authors first applied an unsupervised Gaussian Mixture Model (GMM) to cluster the axonal projections based on their regions of origin and various regions of termination. The clustering results were used to generate synthetic projections, onto which

axonal tufts (terminal arbors) for both local and long-range projections were synthesized using a previously described algorithm that was first developed to generate synthetic dendrites. To validate the projection clustering, modeled connectivity, and algorithm for synthesizing axonal terminal morphologies, the authors provide metrics of the connectivity and morphometrics of 1,695 synthetic pyramidal neurons originating from Layer 5 of the primary motor cortex (MOp5), compared against experimentally reconstructed examples from the aforementioned reconstruction datasets. Additionally, the authors synthesized 1,472 long-range axons based on GMM clusters from other major cortical subregions to compare the accuracy of the modeled connectivity relative to the experimental datasets. In both applications, quantitative comparison of key metrics of both the synthesized connectivity and tuft morphologies showed highly accurate, statistically significant reproduction of the experimentally derived axonal data from the targeted cortical regions.

In summary, the study is thorough, offers a significant contribution, and introduces a highly innovative approach to addressing a complex challenge within the neuroscience community, i.e. creating more complete single-neuron axonal morphologies. This method still faces limitations due to the current availability of fully reconstructed neurons (both dendrites and axons). However, as additional experimental datasets with more training neurons become available, this model/framework has the potential to be highly effective for synthesizing neurons and their projection targets. Thus, with minor revisions, this paper should be a good candidate for publication in Nature Communications.

We thank the reviewer for this summary and his recommendation for publication in Nature Communications.

Minor issues.

- It would be helpful if Figure 1 included a schematic representation of the pipeline, outlining the key steps in developing the algorithm for synthesizing single neurons, the main stages of the automated synthesis process. They should also provide examples of synthesized outputs compared to real examples to illustrate the accuracy of their algorithm.*

We thank the reviewer for this remark. We provided a schematic for the axon synthesis key steps, detailed the methodology, and compared it with biological equivalents in the companion work Ref. 25. In the previous work, we introduced the methodology and the algorithm of axon synthesis in detail. The present work can be seen as an application of the synthesis algorithm, which brings novel aspects such as the Gaussian mixture clustering for selecting targeted regions, as well as new axonal data.

- One of the major components of this work is the synthesis of axonal terminal arbors using an algorithm originally developed to model dendritic processes. Given that axonal terminals are typically much more complex and denser than dendrites, the authors should provide a detailed explanation of their approach and present data to demonstrate the accuracy of the synthesized axonal terminals. This would help validate the robustness of their algorithm for modeling such intricate structures.*

We thank the reviewer for pointing out this aspect of our study. The axon synthesis algorithm was validated in detail in Ref 25. Due to the comments of reviewer #2 for similarity of the two papers we opted to exclude repeating this validation in the current paper. However, due to the theoretical principles of the algorithm, as presented in Kanari et al. 2022 the algorithm

reproduces the topological and morphological properties of the axons within each cluster. The morphometrics of the full axon have been fully validated in Ref 25.

• The authors should clarify whether their algorithm is exclusively suited for the synthesis of cortical pyramidal neurons or whether it can be applied to other neuron types in the brain. If the latter is true, it would be useful to provide examples validating the algorithm's applicability to other brain cell types (e.g., neurons in the striatum, thalamus, or cerebellum).

We thank the reviewer for this comment. In the implementation of the synthesis algorithm, there is no explicit fitting to cortical pyramidal neurons, which implies that the methodology should apply also to other types of neurons. Non pyramidal neurons mainly differ in the dendritic branching. Non pyramidal examples have been provided in Kanari et al. 2022. This work focused on axonal synthesis, therefore the methodology does not differ based on dendritic branching differences. However, our example focused on cortical cells. It happens that projecting axons in the cortex are pyramidal cells and we collected experimental data primarily from the cortex. It is out of scope for the current paper to expand to other regions due to lower number of reconstructions for other regions.

• Similarly, the authors note that their proposed method for synthesizing axons is species-agnostic, suggesting it can be applied to both rats and humans with some parameter tuning. However, it is unclear whether they have demonstrated this approach in species beyond mice. Including examples of the algorithm's application to other species would strengthen their claim.

We thank the reviewer for this comment. Indeed we cannot prove that it would indeed work "out-of-the-box" for other species. However in the whole algorithmic pipeline that we implemented, there is no explicit fitting to the mouse species. The only requirement would be to have data available in the same format, i.e. morphologies and a brain annotation space. It might be needed to adjust the input parameters if one would want to fit morphometrical features on the input data. However, this is out of scope for this study, due to lack of sufficient experimental data. We have modified this statement to clarify this point further in the manuscript.

• Tuft grouping was performed using a maximum clustering radius and path distance of 300 microns. Did the authors test different threshold values before selecting these parameters?

These values were not varied in the present work, but a similar analysis was performed in our methodology paper, Ref 25, Fig 12. We could see that the smaller the clustering distance, the lower the L1 error we had with biological tufts. However the more tufts we have, the more memory and compute intensive. 300 microns seemed a good trade-off between accuracy and compute resources.